# Coarse to Fine Automatic Segmentation of Abdominal Multiple Organs

Yi Lv[1,2][0000−0002−3778−430X], Yu Ning[2][0000−0001−5319−3588], and Junchen Wang[2,3][0000−0002−0916−9932]*

[1] North China Research Institute of Electro-optics, Beijing, 100015, China
[2] School of Mechanical Engineering and Automation, Beihang University, Beijing, 100191, China
[3] Beijing Advanced Innovation Center for Biomedical Engineering, Beihang University, Beijing, China
*Corresponding author (`wangjunchen@buaa.edu.cn`)

**Abstract.** Abdominal multi-organ segmentation is fast becoming a key instrument in preoperative diagnosis. Using the results of abdominal CT image segmentation for three-dimensional reconstruction is an intuitive and accurate method for surgical planning. In this paper, we propose a stable three-stage fast automatic segmentation method for abdominal 13 organs: liver, spleen, pancreas, right kidney, left kidney, stomach, gallbladder, esophagus, aorta, inferior vena cava, right adrenal gland, left adrenal gland, and duodenum. Our method includes preprocessing the CT data, segmenting the multi-organ and post-processing the segmentation outputs. The results on the first-fold validation set show that the average DSC performance on the official validation leaderboard is about 0.77. The average time and GPU memory consumption for each case is 81.42s and 1953MB.

**Keywords:** Medical image segmentation · Deep learning · Neural network.

## 1 Introduction

Abdominal multi organ segmentation is of great significance in medical diagnosis and research. Through pixel level segmentation of CT or MRI and three-dimensional reconstruction of the segmentation results, doctors can obtain more intuitive information of patients' abdominal organs [3,17,4,20,10]. In recent years, medical image automatic segmentation algorithm has made a great breakthrough. Methods based on deep learning has achieved excellent performance in this task [12,9,19,18]. The deep learning technology based on neural networks can achieve fast segmentation, and effectively solve the problem of low accuracy and long time-consuming image segmentation [8,15]. The research in recent years mainly focuses on the network structure and segmentation framework. At present, the most widely used network structure is the encoding-decoding shaped structure similar to U-Net [14], such as 3D U-Net [1] and V-Net [13], and nnU-Net [7] has also achieved excellent results in the field of segmentation framework.

For example, in the MICCAI challenge 2019 kits19 competition, the accuracy of nnU-Net using 3D U-Net in the task of kidney segmentation is very close to that of human, but the required time to complete a segmentation is far less than that of manual segmentation. The deep learning-based methods not only surpass the traditional algorithms, but also approach the accuracy of manual segmentation. However, previous published studies are limited to be used on low-configuration devices.

In this paper, we propose a stable three-stage automatic segmentation method for abdominal 13 organs: liver, spleen, pancreas, right kidney, left kidney, stomach, gallbladder, esophagus, aorta, inferior vena cava, right adrenal gland, left adrenal gland, and duodenum. Our method can complete the segmentation task, including preprocessing the CT data, segmenting the multi-organ and finally post-processing the segmentation outputs, with low GPU memory occupation.

## 2    Methods

### 2.1    Preprocessing

In the preprocessing stage, we first standardize the spacing of CT. Due to the amount of available GPU memory, the patch size that can be processed in 3D CNNs is typically quite limited. Thus, the target spacing, which directly impacts the total size of the images in voxels, also determines how much contextual information the CNN can capture in its patch size. We reshape all the data with the voxel spacing of $4.4 \times 2.5 \times 2.5$ mm for the first step and $3.0 \times 1.6 \times 1.6$ mm for the second step. After spacing standardization, we set the maximum in-plain resolution to $128 \times 176$ pixels for the first step and $230 \times 300$ pixels for the second step, so as to prevent data with high original spacing from being too large after the standardization of spacing and resulting in a significant increase in segmentation time.

### 2.2    Proposed Method

To verify the impact of segmentation pipeline strategy on the results, we used an improved 3D U-Net as the segmentation network. The network architecture is illustrated in Fig. 1. The network includes an encoding path and a decoding path, each of which has four resolution levels. Each layer of the encoding path contains two $3 \times 3 \times 3$ convolution layers, each followed by a ReLu layer, followed by a $2 \times 2 \times$ Maximum pool layer with step size of 2 in each direction of 2. In the decoding path, each layer contains a 2 with a step size of $2 \times 2 \times 2$, followed by two $3 \times 3 \times 3$, each followed by a RuLu layer. The summation between Dice loss and cross entropy loss is chosen as the loss function. We used adaptive moment estimation (Adam) as the optimizer. The batch size was set to be 2. The networks were initialized using Kaiming normal initialization. We set the learning rate to be 1e-3 and reduced the learning rate by a multiplier of 0.99 after every 5 epochs until it reached 1e-6.

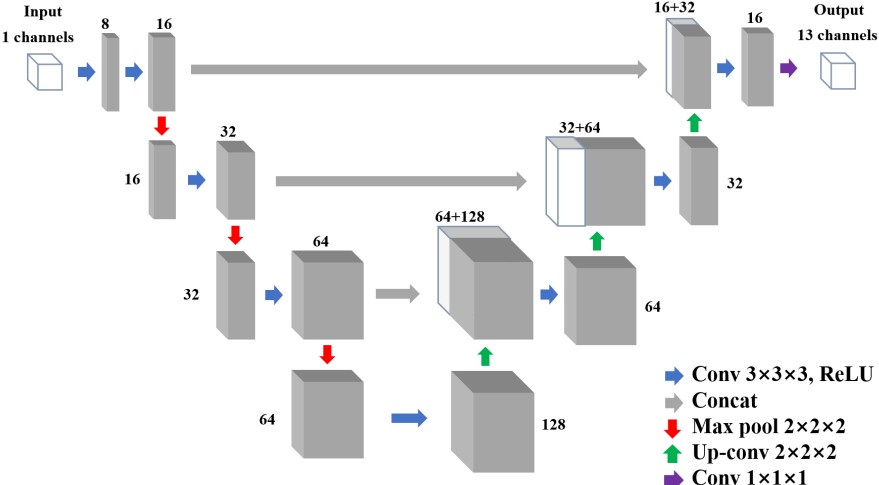

**Fig. 1.** The network architecture. Gray cuboids represent feature maps. The number of channels is denoted next to the feature map.

The pipeline of our method consists of three stages: global locating, organ locating and organ segmentation. Each stage of our method will generate a segmentation result for the complete CT, and the operation of the second and the third steps are based on the previous result. As shown in Fig. 2, in the global locating stage, we first cut the original CT into several ROIs, and then segment each ROI with the first trained neural network. In the organ locating stage, we first locate the region of abdominal organs in the whole CT according to the results of the first step, and then we save this region with a higher resolution and segment it with the second trained network. In the stage of organ segmentation, we locate and crop each organ according to the results of the second step, and then use the corresponding network to fine segment each organ. Finally, we superimpose the segmentation results of each organ to the corresponding position and then generate the feature map of final segmentation result.

In order to further improve the robustness of the network on different data, we adopt the training strategy of semi-supervised learning. In the training process, we use 40 labeled data and 50 unlabeled data as the training used in the stage of global locating and organ locating. We use the labeled data to train the model in the first 50 epochs, and then introduce the unlabeled data.

### 2.3   Post-processing

In the post-processing stage, we splice the results of the network segmentation. We keep the region with the largest volume and remove the rest to eliminate isolated incorrectly predicted labels. To improve the segmentation efficiency of our method, we clear the cache and delete the used feature map and the model

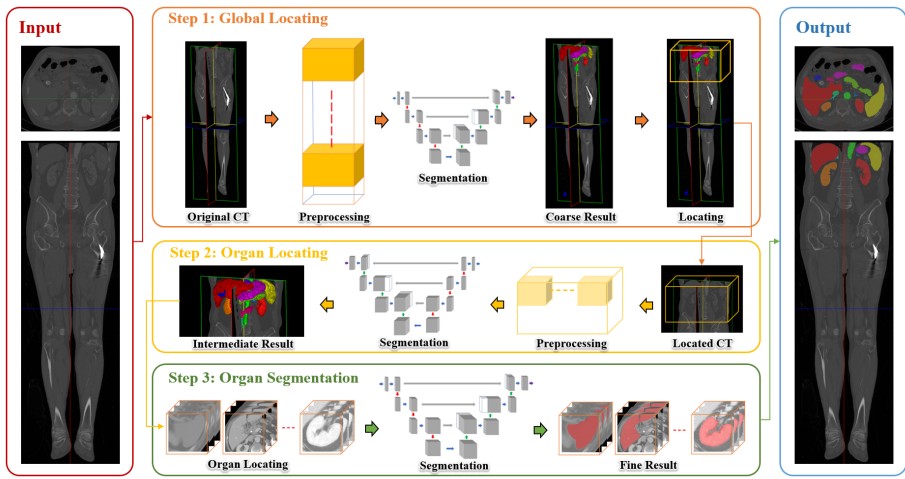

**Fig. 2.** The pipeline of our proposed method.

from the GPU after each step. Finally, the maximum GPU memory we use is 1953MB.

## 3    Experiments

### 3.1    Dataset and evaluation measures

The FLARE2022 dataset is curated from more than 20 medical groups under the license permission, including MSD [16], KiTS [5,6], AbdomenCT-1K [11], and TCIA [2]. The training set includes 50 labelled CT scans with pancreas disease and 2000 unlabelled CT scans with liver, kidney, spleen, or pancreas diseases. The validation set includes 50 CT scans with liver, kidney, spleen, or pancreas diseases. The testing set includes 200 CT scans where 100 cases has liver, kidney, spleen, or pancreas diseases and the other 100 cases has uterine corpus endometrial, urothelial bladder, stomach, sarcomas, or ovarian diseases. All the CT scans only have image information and the center information is not available.

The evaluation measures consist of two accuracy measures: Dice Similarity Coefficient (DSC) and Normalized Surface Dice (NSD), and three running efficiency measures: running time, area under GPU memory-time curve, and area under CPU utilization-time curve. All measures will be used to compute the ranking. Moreover, the GPU memory consumption has a 2 GB tolerance.

### 3.2    Implementation details

**Environment settings** The environments and requirements are presented in Table 1.

**Table 1.** Environments and requirements.

| | |
|---|---|
| **Windows/Ubuntu version** | Windows 10 |
| **CPU** | AMD Ryzen 7 5800X |
| **RAM** | 8GB × 4 |
| **GPU (number and type)** | One RTX8000 48G |
| **CUDA version** | 10.2 |
| **Programming language** | Python 3.7.9 |
| **Deep learning framework** | Pytorch(Torch 1.8.0, torchvision 0.9.0) |
| **Average inference time** | 81.42s |
| **GPU memory consumption** | 1953 MB |

**Training protocols** The Training protocols are presented in Table 2.

**Table 2.** Training protocols.

| | |
|---|---|
| **Network initialization** | Kaiming normal initialization |
| **Data augmentation methods** | Scaling, rotations, brightness, contrast, gamma |
| **Batch size** | 8 |
| **Patch size** | $64 \times 128 \times 176$ |
| **Total epochs** | 100 |
| **Optimizer** | Adam |
| **Initial learning rate (lr)** | 0.01 |
| **Lr decay schedule** | multiplied by 0.99 every 5 epochs |
| **Training time** | 5.9 hours |
| **Number of model parameters** | 1.328M |
| **Number of flops** | 33.263 G |
| **Loss function** | Combination of Dice loss and WCE loss |

### 3.3   Resource consumption

The Resource consumption during inference is presented in Table 3.

**Table 3.** Resource consumption during inference.

| | |
|---|---|
| **Total Running Time on Validation Set** | 67.85 mins |
| **Maximum RAM consumption** | < 8 GB |
| **Maximum GPU memory consumption** | 1953 MB |

## 4    Results and discussion

As the accuracy metrics, the average DSC between the predicted mask and the ground truth mask were employed. Assume $A$ and $B$ are two masks, the metric is given by (1).

$$\text{DSC} = \frac{2(A \cap B)}{A + B} \tag{1}$$

### 4.1    Quantitative results on validation set

Table 4 compares the experimental data on the segmentation results on 13 organs in the three stages. In the stage of global locating, organ locating and organ segmentation, our method achieves average DSC of 0.63, 0.73 and 0.77 respectively. The highest DSC between the three stages are highlighted in Table 4. It is apparent from this table that the DSC results in stage 3 is significantly higher than the previous stages.

**Table 4.** Comparison on 13 Structures of official validation.

|  | Stage 1 | Stage 2 | Stage 3 |
|---|---|---|---|
| Liver | 0.902 | 0.868 | **0.903** |
| Right Kidney | 0.798 | 0.864 | **0.896** |
| Spleen | 0.802 | 0.898 | **0.926** |
| Pancreas | 0.497 | **0.708** | 0.683 |
| Aorta | 0.837 | 0.911 | **0.930** |
| Inferior Vena Cava | 0.710 | 0.802 | **0.846** |
| Right Adrenal Gland | 0.490 | 0.600 | **0.653** |
| Left Adrenal Gland | 0.354 | 0.528 | **0.610** |
| Gallbladder | 0.393 | 0.463 | **0.562** |
| Esophagus | 0.573 | 0.652 | **0.671** |
| Stomach | 0.672 | 0.780 | **0.799** |
| Duodenum | 0.379 | 0.577 | **0.593** |
| Left Kidney | 0.789 | 0.856 | **0.867** |
| Average | 0.630 | 0.731 | **0.765** |

As the models used in the first and the second stage were semi-supervised trained with unlabeled data, we also test the effect of unlabeled data. Table 5 shows the DSC comparison of our method with and without using unlabeled data. It can be observed that the accuracy of our method using unlabeled data has been improved.

### 4.2    Qualitative results on validation set

Figure 3 shows three examples with good segmentation results on CT slices in validation set. Figure 4 shows the results with voxel-based rendering from three

**Table 5.** Comparison of our method with and without using unlabeled data.

|                        | Average DSC | Standard Deviation of DSC |
| ---------------------- | ----------- | ------------------------- |
| With Unlabeled Data    | 0.731       | 0.142                     |
| Without Unlabeled Data | 0.678       | 0.186                     |

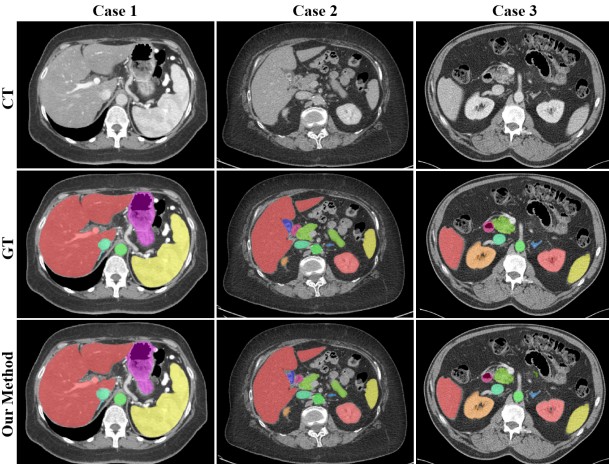

**Fig. 3.** Three examples with good segmentation results on CT slices.

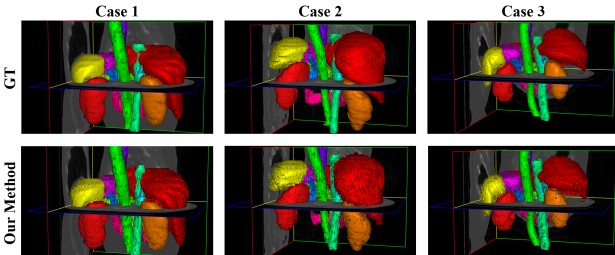

**Fig. 4.** Segmentation results with voxel-based rendering from three examples in the evaluation dataset. For each example, the ground truth and the segmentation results are given for visual comparison.

examples in the validation set. In these results, the performance of our method is generally stable.

As shown in Figure 5, there also have examples with bad segmentation results on CT slices in validation set. In the first case of the bad results, part of the right kidney tumor and pancreas were not correctly recognized. This is because there is not much data with kidney tumors in the training set, and the characteristic boundary between pancreas and surrounding tissues is not particularly obvious. In the second case, our method performs bad on spleen and stomach. The gray value of stomach is abnormally high in CT image, which not only led to the wrong recognition of the stomach, but also covered the correct label of spleen. In the third case, a typical liver recognition error occurred. Due to the rarity of such features in training data, the network habitually takes the lung boundary as the criterion for judging the region of liver.

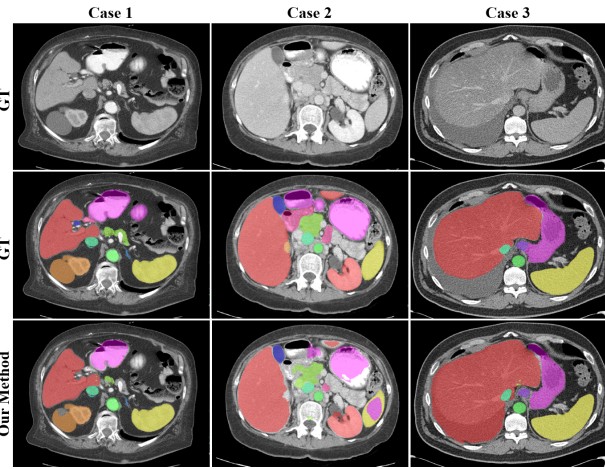

**Fig. 5.** Three examples with bad segmentation results on CT slices.

## 5   Conclusion

We propose a three-stage automatic segmentation method for abdominal 13 organs based on improved 3D U-Net. The results show that the average dice of our method is 0.77 on the official validation leaderboard. The results show that the accuracy of our method on massive organs is better than that for small organs. The speed of three-stage method is fast, but it is difficult to achieve higher accuracy due to the limitation of feature map size. Future work will focus on promoting accuracy based on less stage methods, in which the segmentation speed can be further improved.

**Acknowledgements** The authors of this paper declare that the segmentation method they implemented for participation in the FLARE 2022 challenge has not used any pre-trained models nor additional datasets other than those provided by the organizers. The proposed solution is fully automatic without any manual intervention. This work was supported by Natural Science Foundation of China (Grant No. 62173014) and Natural Science Foundation of Beijing Municipality (Grant No. L192057).

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
