# OpenReview forum: "Coarse to Fine Automatic Segmentation of Abdominal Multiple Organs"
_MICCAI.org/2022/Challenge/FLARE_

### Official Review · Reviewer_Kaqt · 2022-09-16
**Good work, the inference time is acceptable and the GPU memory consumption is really low！**

**Rating:** 9
**Confidence:** 4

**Review:**

Pros：
1）In this work, they do segmentation by three stages: global locating, organ locating and organ segmentation which reducing the resource consumption a lot.
2) The final mean DSC is 0.77 and the average inference time is 81.42s.
3) The RAM usage and GPU consumption are pretty low

Cons:
1) The way to use the unlabeled data is talked little.

---

### Meta-Review · Program_Chairs · 2022-09-28

**Recommendation:** Major Revision
**Confidence:** 5

**Metareview:**

Please address the three reviewers' comments here
https://openreview.net/forum?id=WgYphAbJZS-&referrer=%5BProgram%20Chair%20Console%5D(%2Fgroup%3Fid%3DMICCAI.org%2F2022%2FChallenge%2FFLARE%2FProgram_Chairs%23paper-status)